# IBADR: an Iterative Bias-Aware Dataset Refinement Framework for Debiasing NLU models

**Xiaoyue Wang**[a,c,†] **Xin Liu**[a,c,†] **Lijie Wang**[b], **Yaoxiang Wang**[a,c], **Jinsong Su**[a,c,*] and **Hua Wu**[b]

[a]School of Informatics, Xiamen University, Xiamen 361005, China,

[b]Baidu Inc., Beijing 100085, China,

[c]Key Laboratory of Digital Protection and Intelligent Processing of Intangible Cultural Heritage of Fujian and Taiwan (Xiamen University), Ministry of Culture and Tourism, China

{xiaoyuewang, liuxin}@stu.xmu.edu.cn, jssu@xmu.edu.cn

## Abstract

As commonly-used methods for debiasing natural language understanding (NLU) models, dataset refinement approaches heavily rely on manual data analysis, and thus maybe unable to cover all the potential biased features. In this paper, we propose IBADR, an Iterative Bias-Aware Dataset Refinement framework, which debiases NLU models without predefining biased features. We maintain an iteratively expanded sample pool. Specifically, at each iteration, we first train a shallow model to quantify the bias degree of samples in the pool. Then, we pair each sample with a bias indicator representing its bias degree, and use these extended samples to train a sample generator. In this way, this generator can effectively learn the correspondence relationship between bias indicators and samples. Furthermore, we employ the generator to produce pseudo samples with fewer biased features by feeding specific bias indicators. Finally, we incorporate the generated pseudo samples into the pool. Experimental results and in-depth analyses on two NLU tasks show that IBADR not only significantly outperforms existing dataset refinement approaches, achieving SOTA, but also is compatible with model-centric methods. [1]

## 1 Introduction

Although neural models have made significant progress in many natural language understanding (NLU) tasks (Bowman et al., 2015; Gururangan et al., 2018), recent studies have demonstrated that these models exhibit limited generalization to out-of-distribution data and are vulnerable to various types of adversarial attacks (Dasgupta et al., 2018; McCoy et al., 2019). This is primarily due to their tendency to rely excessively on biased features—spurious surface patterns that are falsely associated with target labels, rather than to capture the underlying semantics. Consequently, how to effectively debias neural networks has become a prominent research topic, attracting increasing attention recently.

To alleviate this issue, researchers have proposed many methods that can be generally divided into two categories: *model-centric mitigation approaches* (Clark et al., 2019; Stacey et al., 2020; Utama et al., 2020a; Karimi Mahabadi et al., 2020; Du et al., 2021) and *dataset refinement approaches* (Lee et al., 2021; Wu et al., 2022; Ross et al., 2022). The former mainly focuses on designing model architectures or training objectives to prevent models from exploiting dataset biases, and the latter aims to adjust dataset distributions to reduce correlations between spurious features and labels. In these two types of methods, dataset refinement approaches possess the advantage of not requiring modifications to the model architecture and training objective, while are also compatible with model-centric approaches. Therefore, in this work, we also concentrate on dataset refinement approaches, which employ controllable generation techniques (Zhou et al., 2020; Hu et al., 2022) to refine the data distribution. However, recent studies (Lee et al., 2021; Wu et al., 2022; Ross et al., 2022) heavily rely on manual data analysis for debiasing models. They either define perturbation rules to generate adversarial examples, or generate pseudo samples and filter out samples with identified biased features. Typically, the state-of-the-art (SOTA) method (Wu et al., 2022) first generates a large amount of samples and then applies *z-filtering* involving a predefined set of biased features to eliminate samples with such features. However, these methods of manually predefining biased features may overlook some potential biased features, thus limiting their generalizability.

In this paper, we propose **IBADR**, an Iterative Bias-Aware Dataset Refinement framework, which

---

[1]We release our code at http://github.com/DeepLearnXMU/IBADR.

[†]These authors contributed equally to this work.

[*]Corresponding author.

iteratively generates samples to debias NLU models without predefining biased features. Under this framework, we create a sample pool initialized by the original training samples, and gradually expand it through multiple iterations. As shown in Figure 1, in each iteration, we first sort and group samples in the pool according to their bias degree, determined by a shallow model trained on a limited set of training samples. Next, we concatenate samples in each group with a *bias indicator* that represents its bias degree. These concatenated samples are then utilized to train a sample generator, which effectively learns the correspondence relationship between bias indicators and samples. Afterwards, as implemented in the training phase, we feed a low-degree bias indicator to the sample generator, allowing it to generate pseudo samples with fewer biased features. Finally, we add these pseudo samples back into the sample pool and repeat the above process until the maximum number of iterations is reached.

Apparently, the above iterative process guides the sample generator towards samples with fewer biased features. However, we observe the generated pseudo samples display less diversity when we feed the lowest-degree bias indicator to the sample generator. The underlying reason is that the shallow model consistently assigns a relatively low bias degree to samples with specific patterns, such as the premise directly negating the hypothesis by inserting a word "*not*". Consequently, the sample generator learns these patterns and tends to produce samples containing similar patterns, thereby limiting their diversity.

To address this issue, we further explore two strategies to diversify generations. First, instead of always using the lowest-degree bias indicator, we randomly select a low-degree bias indicator. In this way, the sample generator is discouraged from continually creating pseudo samples containing similar patterns, while still ensuring fewer biased features in the pseudo samples. Secondly, we dynamically update the shallow model by integrating the newly generated pseudo samples during the iterative generation process. By doing this, we effectively decrease the assignment of the lowest-degree bias indicator to pattern-specific samples, ultimately promoting greater diversity of the generated samples.

To summarize, the main contributions of this paper are three-fold:

- We propose a dataset refinement framework designed to iteratively generate pseudo samples without prior analysis of biased features.

- We present two strategies to enhance the diversity of the pseudo samples, which further boost the performance of NLU models.

- To verify the effectiveness and generality of IBADR, we conduct experiments on two NLU tasks. The experimental results show that IBADR achieves SOTA performance.

## 2 The IBADR Framework

In this section, we give a detailed description of IBADR. Under this framework, we first use a limited set of training samples to train a shallow model, which serves to measure the bias degree of samples. Then, we iteratively generate pseudo samples with fewer biased features, as illustrated in Figure 1. Finally, these pseudo samples are used to debias the NLU models via retraining.

### 2.1 Training a Shallow Model to Measure the Bias Degree of Samples

As investigated in (Utama et al., 2020b), a shallow model trained on a small portion of training data tends to overfit on biased features, thus is highly confident on the samples that contain biased features. Motivated by this, we randomly select some training samples to train a shallow model, denoted as $\theta_s$, for measuring the bias degree of samples.

Let $(x^{(i)}, y^{(i)})$ denote a training sample for NLU tasks, where $y^{(i)}$ is the golden label of the input $x^{(i)}$, we directly use the model confidence $p(y^{(i)}|x^{(i)}; \theta_s)$ to quantify the bias degree of $(x^{(i)}, y^{(i)})$. Apparently, if $p(y^{(i)}|x^{(i)}; \theta_s) \to 1$, $(x^{(i)}, y^{(i)})$ is more likely to be a biased one.

Back to our framework, our primary objective is to generate samples with a low bias degree, which can be used to reduce spurious correlations via adjusting dataset distributions.

### 2.2 Iterative Pseudo Sample Generation

The overview of the iterative sample generation process is shown in Figure 1. During this process, we introduce a sample generator $\theta_g$ to iteratively generate pseudo samples, which are added into a sample pool $\mathcal{S}$. Specifically, we initialize the sample pool $\mathcal{S}$ with the original training samples, the sample generator $\theta_g$ with a generative pretrained

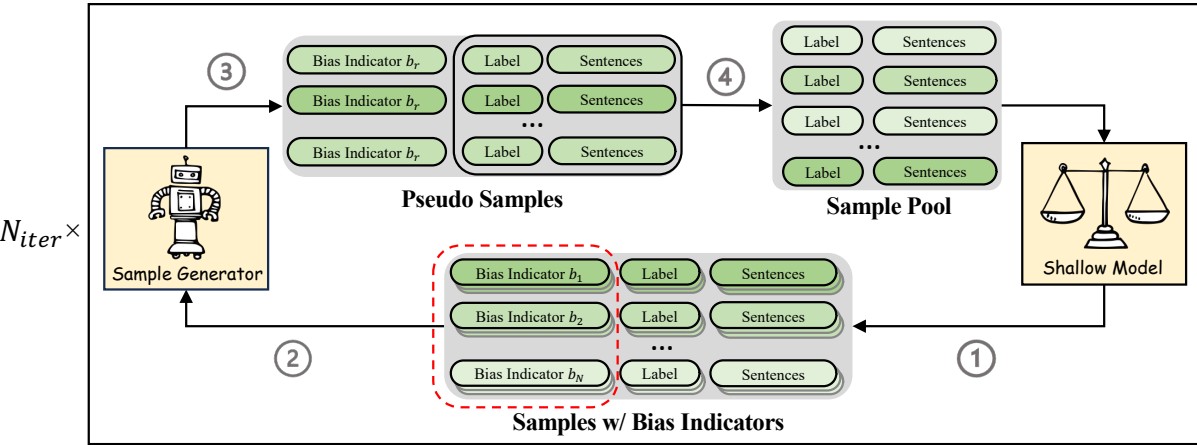

Figure 1: Overview of the iterative sample generation process, which consists of four key stages: ① Setting bias indicator; ② Finetuning sample generator; ③ Generating pseudo samples; and ④ Expanding sample pool. Through $N_{iter}$ iterations of the above steps, we continuously augmented the sample pool with pseudo samples, which can be effectively employed to debias the NLU models.

language model. Then, we iteratively expand $\mathcal{S}$ via the following four stage:

**Step 1: Setting Bias Indicators.** First, we use the above-mentioned shallow model to measure the bias degree of each sample in $\mathcal{S}$, as described in Section 2.1, and sort these samples according to their bias degree and divide them into $N_{bi}$ groups with equal size. Each group is assigned with a bias indicator $b_n$, where $1 \leq n \leq N_{bi}$, $b_1$ represents the lowest-degree bias indicator and $b_{N_{bi}}$ denotes the highest-degree bias indicator.

**Step 2: Finetuning Sample Generator.** Then, we use the samples in $\mathcal{S}$ to finetune the sample generator $\theta_g$ via the following loss function:

$$\mathcal{L}_g = -\sum_i^{|\mathcal{S}|} \log p(x^{(i)}|b^{(i)}, y^{(i)}; \theta_g), \quad (1)$$

where $b^{(i)}$ represents the bias indicator assigned for the training sample $(x^{(i)}, y^{(i)})$. Through training with this objective, the generator can effectively learn the correspondence relationship between bias indicators and samples. Furthermore, in the subsequent stages, we can specify both the bias indicator and the label to control the generations of pseudo samples.

**Step 3: Generating Pseudo Samples.** Next, we designate a bias indicator $\bar{b}$ representing a low degree of bias, and then feed it with a randomly-selected NLI label $\bar{y}$ into the generator $\theta_g$. This process allows us to form a pseudo sample $(\bar{x}, \bar{y})$ by sampling $\bar{x}$ from the generator output distribution

$p_g(\cdot|\bar{b}, \bar{y}; \theta_g)$. By repeating this sampling process, we can obtain a set of generated pseudo samples with fewer biased features.

**Step 4: Expanding Sample Pool.** Subsequently, to ensure the quality of generated pseudo samples, we follow Wu et al. (2022) to filter the above generated pseudo samples with model confidence lower than a threshold $\epsilon$, and incorporate the remaining pseudo samples back into $\mathcal{S}$.

After $N_{iter}$ iterations of the above steps, our sample pool contains not only the original training samples, but also abundant pseudo samples with fewer biased features. Finally, we debias the NLU model via the retraining on these samples.

### 2.3 Diversifying Pseudo Samples

Intuitively, the most direct way is to set the above specified bias indicator $\bar{b}$ to $b_1$, which denotes the lowest bias degree. However, we observe that such generated pseudo samples lack diversity and fail to cover diverse biased features. The reason behind this is that the generated pseudo samples designated with $b_1$ always follow certain patterns, exhibiting less diversity compared to those assigned with other bias indicators. For example, the premise directly negates the hypothesis using the word "*not*". Consequently, this results in spurious correlations between $b_1$ and these certain patterns. Hence, the generator tends to generate samples following these patterns and fails to generate samples that compass a broader range of biased features.

To address this issue, we employ the following two strategies: (i) Instead of using the lowest-

degree bias indicator $b_1$, we use a randomly-selected low-degree bias indicator: $\bar{b}=b_r$, where $1 \leq r \leq \frac{N_{bi}}{2}$, and feed it into the generator during the iterative generation process. Upon human inspection, we observe that the generated pseudo samples not only become diverse but also still contain relatively few biased features. (ii) During the generation process, we update the shallow model $\theta_s$ using a randomly-extracted portion of $\mathcal{S}$ at each iteration. This strategy prevents the shallow model from consistently predicting a low bias degree to pseudo samples following previously-appeared patterns, thereby enhancing the diversity of the pseudo samples.

# 3 Experiments

## 3.1 Setup

**Tasks and Datasets.** We conduct experiments on two NLU tasks: natural language inference and fact verification.

- **Natural Language Inference (NLI).** This task aims to predict the entailment relationship between the pair of premise and hypothesis. We conduct experiments using the MNLI (Williams et al., 2018) and SNLI (Bowman et al., 2015) datasets. As conducted in previous studies (Stacey et al., 2020; Wu et al., 2022; Lyu et al., 2023), in addition to the development sets, we evaluate IBADR on the corresponding challenge sets for MNLI and SNLI, namely HANS (McCoy et al., 2019) and the Scramble Test (Dasgupta et al., 2018), respectively. These two challenge sets are specifically designed to assess whether the model relies on syntactic and word-overlap biases to make predictions.

- **Fact Verification.** This task is designed to determine whether a textual claim is supported or refuted by the provided evidence text. We select FEVER (Thorne et al., 2018) as our original dataset and evaluate the model performance on the development set and two challenge sets: FeverSymmetric V1 and V2 (Symm.v1 and Symm.v2) (Schuster et al., 2019a), both of which are developed to mitigate biases stemming from claim-only data.

**Baselines.** We compare IBADR with the following baselines:

| Dataset | Original | Augmented |
|---------|----------|-----------|
| MNLI | 393K | 1.0M |
| SNLI | 549K | 1.1M |
| FEVER | 243K | 607K |

Table 1: Sample numbers of the constructed augmented datasets for MNLI, SNLI, and FEVER.

| $N_{bi}$ | MNLI (Acc.) | |
|----------|-------------|-------------|
| | dev-m | dev-mm |
| 3 | 85.03 | 85.63 |
| 5 | **85.31** | **85.88** |
| 7 | 84.64 | 84.35 |
| 9 | 84.47 | 84.19 |

Table 2: Results on the development sets of MNLI with different numbers of bias indicators.

- **CrossAug** (Lee et al., 2021). This method tackles negation bias in the fact verification task through a contrastive data augmentation method.

- **z-filter** (Wu et al., 2022). It first defines a set of task-relevant biased features, and then trains a generator on existing datasets to generate pseudo samples, where pseudo samples with these biased features are filtered. Finally, the remaining samples are used to retrain the model.

- **Products-of-Experts (PoE)** (He et al., 2019; Karimi Mahabadi et al., 2020). In an ensemble manner, it trains a debiased model with a bias-only one, the predictions of which heavily rely on biased features. By doing so, the debiased model is encouraged to focus on samples with fewer biased features where the bias-only model performs poorly.

- **Confidence Regularization (Conf-reg)** (Utama et al., 2020a). This method trains a debiased model by increasing the uncertainty of samples with biased features. It first trains a bias-only model to quantify the bias degree of each sample, and then scales the output distribution of a teacher model based on the bias degree, where the re-scaled distribution can be used to enhance the debiased model.

- **Example Reweighting (Reweight)** (Schuster et al., 2019b). This method aims to reduce the contribution of samples with biased features

| Method | MNLI (Acc.) | | | SNLI (Acc.) | | FEVER (Acc.) | | |
|---|---|---|---|---|---|---|---|---|
| | dev-m | dev-mm | HANS | dev | Scramble | dev | Symm.v1 | Symm.v2 |
| BERT-base | 83.87 | 84.11 | 61.22 | 90.61 | 72.74 | 87.06 | 56.53 | 63.84 |
| Reweight (Schuster et al., 2019b)* | 82.56 | – | 66.18 | 86.44 | 80.30 | 83.45 | 61.56 | 67.33 |
| Conf-reg (Utama et al., 2020a)* | 84.60 | 85.00 | 69.10 | 90.56 | 83.21 | 85.31 | 59.69 | 64.75 |
| DCT (Lyu et al., 2023)* | 84.19 | – | 68.30 | **90.64** | 86.40 | 87.12 | 63.27 | 68.45 |
| CrossAug (Lee et al., 2021)* | – | – | – | – | – | 85.34 | 68.90 | – |
| z-filter (Wu et al., 2022)* | 82.55 | 82.70 | 67.69 | 88.08 | 87.87 | – | – | – |
| IBADR | **85.17**† | **85.05**† | **71.67**† | 90.32 | **92.31**† | **89.61**† | **70.01**† | **72.61**† |

Table 3: Results on the development and challenge sets (HANS, Scramble, Symm.v1, Symm.v2) of MNLI, SNLI and FEVER. * means the results are directly cited from previous studies. Note that IBADR outperforms all baselines on the challenge sets, while maintaining comparable or better performance on the development sets. † indicates the results are significantly better than the best comparison method ($p < 0.001$).

on the training loss by assigning them with relatively small weights.

- **Debiasing Contrastive Learning (DCT)** (Lyu et al., 2023). It applies contrastive learning to mitigate biased latent features by utilizing a specifically designed sampling strategy.

Please note that in exception to CrossAug and z-filter, which are dataset refinement approaches, all other approaches are model-centric.

**Implementation Details.** In our experiments, we use GPT2-large (Radford et al., 2019) to construct the sample generator and the shallow model, respectively. To train the shallow models for different NLU tasks, we randomly select 2K, 2K, and 0.5K samples from the original training sets of MNLI, SNLI, and FEVER, individually. These shallow models are trained for 3 epochs with a learning rate of 5e-5. When training the sample generator, we set the learning rate to 5e-5, the number of pseudo sample generated per iteration to 200K, and the iteration number $N_{iter}$ to 5. Particularly, we train the sample generator for 3 epochs in the first iteration and for only 1 epoch in the subsequent iterations. When updating the shallow model, we randomly select 2K samples from the sample pool to finetune it.

For the NLU models, we train them on the augmented datasets of different tasks for 8 epochs using a learning rate of 1e-5. We employ an early stop strategy during the training process. We conduct all experiments three times, each time with different random seeds, and report the average results. Sample numbers of the augmented datasets are listed in Table 1.

## 3.2 Effect of Bias Indicator Number $N_{bi}$

The bias indicator number $N_{bi}$ on our framework is an important hyper-parameter, which determines the partition granularity of the sample pool. Thus, we gradually vary $N_{bi}$ from 3 to 9 with an increment of 2 in each step, and compare the model performance on the development sets of MNLI.

As shown in Table 2, when $N_{bi}$ is set to a smaller value, such as 3, there is a significant decrease in model performance. This is because in this case, the bias indicator can only be set to $b_1$, which reduces the diversity of generated pseudo samples, as discussed in Section 2.3. Conversely, when $N_{bi}$ is set to larger values, e.g., 7 or 9, the model performance on both development sets also decreases. We hypothesize that this decline occurs because a larger value of $N_{bi}$ results in a fine-grained partition of the sample pool, reducing the number of samples corresponding to each specific bias indicator. Consequently, this weakens the correspondence relationship between bias indicators and samples, and thus harms the performance of the sample generator. According to these results, we set $N_{bi}$ to 5 for all subsequent experiments.

## 3.3 Main Results

Table 3 presents the experimental results. Overall, compared with all baselines, IBADR is able to achieve the most significant improvements on the challenge sets (i.e. HANS, Scramble, Symm.v1, and Symm.v2). Specifically, IBADR achieves improvements of 2.57, 4.44, 1.11 and 4.16 points than the previously-reported best results, respectively. Note that IBADR is effective on both the development and challenge sets of MNLI and FEVER,

| Method | MNLI (Acc.) | | | SNLI (Acc.) | | FEVER (Acc.) | | |
|---|---|---|---|---|---|---|---|---|
| | dev-m | dev-mm | HANS | dev | Scramble | dev | Symm.v1 | Symm.v2 |
| PoE (Karimi Mahabadi et al., 2020)* | 84.58 | 84.85 | 66.31 | 83.69 | 79.51 | 82.23 | 62.19 | 67.36 |
| z-filter+PoE (Wu et al., 2022)* | 85.22 | 85.72 | 68.75 | – | – | – | – | – |
| IBADR | **85.31** | **85.88** | 71.67 | 90.32 | 92.31 | **89.61** | 70.01 | 72.61 |
| IBADR+PoE | 85.11 | 85.07 | **73.80** | **90.79** | **94.76** | 89.23 | **70.99** | **73.10** |

Table 4: Results on the development and challenge sets of MNLI, SNLI, and FEVER. The combination of IBADR with PoE can significantly enhance the model performance on the challenge sets, surpassing all baselines.

| Method | MNLI (Acc.) | | |
|---|---|---|---|
| | dev-m | dev-mm | HANS |
| IBADR | **85.31** | **85.88** | **71.68** |
| w/o USM | 84.08 | 83.94 | 68.90 |
| $b_r \Rightarrow b_1$ | 84.52 | 84.53 | 69.45 |
| w/o USM & $b_r \Rightarrow b_1$ | 83.76 | 83.61 | 67.79 |
| w/o bias indicator | 85.21 | 84.89 | 63.07 |
| w/o iterative generation | 84.54 | 84.79 | 66.97 |

Table 5: Ablation study on MNLI.

while other baselines, for example, Reweight and z-filter, decline on the development sets.

## 3.4 Compatibility of IBADR with PoE

To assess the compatibility of IBADR with model-centric debiasing methods, we report the model performance when simultaneously using IBADR and PoE (He et al., 2019), following the setting of Wu et al. (2022).

As shown in Table 4, on the challenge sets, the combination of IBADR and PoE not only yields better results than using PoE or IBADR individually, but also outperforms the combination of z-filter and PoE. Thus, we believe that IBADR has the potential to further enhance the performance of existing model-centric methods.

## 3.5 Ablation Study

To assess the effects of special designs on IBADR, we also report the performance of several IBADR variants on MNLI:

- *w/o USM*. In this variant, we do not Update the Shallow Model during the process of iterative sample generation.

- $b_r \Rightarrow b_1$. The sample generator uses the lowest bias indicator $b_1$ rather than the randomly-selected low-degree bias indicator $b_r$, where $1 \leq r \leq \frac{N_{bi}}{2}$, to generate pseudo samples.

- *w/o USM & $b_r \Rightarrow b_1$*. In this variant, the sample generator uses the bias indicator $b_1$ to generate pseudo samples, and the shallow model remains fixed during the generation process.

- *w/o bias indicator*. This variant directly uses the samples without the bias indicator to train the sample generator.

- *w/o iterative generation*. Instead of generating pseudo samples iteratively, we only utilize the sample generator trained in the first iteration to generate pseudo samples.

As shown in Table 5, all variants exhibit performance declines on HANS, indicating the effectiveness of our special designs. Particularly, *w/o bias indicator* demonstrates the most significant performance drop, which is intuitive since the bias indicator can guide the sample generator to produce pseudo samples with fewer biased features. Without the bias indicators, the generated pseudo samples will contain undesired biased features, resulting in poorer performance on the challenge set.

## 3.6 Adversarial Tests for Combating Distinct Biases in NLI

As shown in (Liu et al., 2020), current debiasing approaches primarily concentrate on addressing known biases, and thus might fail to mitigate unknown biases in NLI tasks. To assess the robustness of NLI models, Liu et al. (2020) introduce several comprehensive test sets to evaluate the model performance across various types of biased features, including partial input heuristics (**PI**), inter-sentence heuristics (**IS**), logical inference ability (**LI**), and stress tests (**ST**). Moreover, they propose several data augmentation strategies to improve

| Method | PI-CD | PI-SP | IS-SD | IS-CS | LI-LI | LI-TS | ST | Avg. |
|--------|-------|-------|-------|-------|-------|-------|-------|------|
| Text Swapping* | 71.7 | 72.8 | 63.5 | 67.4 | 86.3 | **86.8** | 66.5 | 73.6 |
| Sub(synonym)* | 69.8 | 72.0 | 62.4 | 65.8 | 85.2 | 82.8 | 64.3 | 71.8 |
| Sub(MLM)* | 71.0 | 72.8 | 64.4 | 65.9 | 85.6 | 83.3 | 64.9 | 72.6 |
| Paraphrasing* | 72.1 | 74.6 | 66.5 | 66.4 | 85.7 | 83.1 | 64.8 | 73.3 |
| BERT-base(MNLI) | 70.3 | 73.7 | 53.5 | 64.8 | 85.5 | 81.6 | 69.2 | 71.2 |
| IBADR | **72.49** | **76.28** | **71.67** | **68.75** | **91.43** | 82.46 | **71.90** | **76.43** |

Table 6: Results on the NLI adversarial test benchmark (Liu et al., 2020). We compare IBADR with the data augmentation techniques investigated by Liu et al. (2020). BERT-base(MNLI) indicates the BERT-base model trained on the training data of MNLI. Note that training on the IBADR's augmented datasets significantly improves the model performance on nearly all test sets.

the model generalization: (i) **Text Swapping**: this strategy exchanges the premise and hypothesis in each original training sample; (ii) **Sub(synonym)**: words in the hypothesis are randomly replaced with synonyms; (iii) **Sub(MLM)**: a masked language model is used to predict the randomly masked words in the hypothesis; (iv) **Paraphrasing**: the hypotheses are paraphrased through back translation.

We compare IBADR with the above data augmentation strategies in Table 6. Overall, IBADR achieves better results on nearly all test sets. This reveals that IBADR can effectively mitigate various biases simultaneously. We attribute this success to the fact that IBADR does not rely on predefined biased features, which enables it to better handle unknown biases.

## 3.7 Generalization on Different Sizes of Pre-trained Language Models

To further explore the compatibility of IBADR with different sizes of pre-trained language models, we reconduct experiments by individually replacing the BERT-base model with BERT-large, RoBERTa-base and RoBERTa-large. We also compare IBADR with $z$-filter, which is the current SOTA data refinement method. The comparisons are performed on the MNLI and SNLI datasets.[†]

As presented in Table 7, IBADR consistently outperforms $z$-filter on all test datasets. When respectively using BERT-large, RoBERTa-base, and RoBERTa-large as pretrained models, IBADR achieves average improvements of 3.29, 1.88, and

---

[†]Since the code of $z$-filter only supports MNLI and SNLI, we exclusively compare our results with theirs for these two datasets.

| | Test data | Original | $z$-filter | Ours |
|---|-----------|----------|-----------|------|
| BERT-large | MNLI dev-m | 86.65 | 85.29 | **86.83** |
| | MNLI dev-mm | 85.91 | 84.70 | **86.62** |
| | HANS | 65.28 | 70.35 | **77.22** |
| | SNLI dev | **91.74** | 88.71 | 91.22 |
| | Scramble | 81.41 | 85.77 | **91.99** |
| | Adv.Test | 76.66 | 78.10 | **78.71** |
| RoBERTa-base | MNLI dev-m | 87.20 | 85.93 | **87.68** |
| | MNLI dev-mm | 87.49 | 86.25 | **87.92** |
| | HANS | 72.19 | 75.55 | **76.26** |
| | SNLI dev | **91.90** | 88.24 | 91.80 |
| | Scramble | 83.07 | 90.01 | **93.21** |
| | Adv.Test | 78.52 | 79.67 | **80.03** |
| RoBERTa-large | MNLI dev-m | 89.80 | 88.38 | **90.11** |
| | MNLI dev-mm | 90.03 | 88.89 | **90.14** |
| | HANS | 76.35 | 77.92 | **81.74** |
| | SNLI dev | **92.87** | 88.30 | 92.70 |
| | Scramble | 85.22 | 87.43 | **95.31** |
| | Adv.Test | 82.22 | 81.97 | **82.95** |

Table 7: Results on MNLI and SNLI when training different sizes of models on augmented datasets of $z$-filter and IBADR.

3.34 point, compared to $z$-filter. These results clearly demonstrate the superior generalizability of IBADR across different sizes of pre-trained language models.

## 3.8 Results on Different Sizes of Datasets

To investigate the robustness of IBADR when the number of original training samples is limited, we

| Data Size | Test Data | BERT-base | Ours |
|---|---|---|---|
| 100K | dev-m | 81.03 | **84.38** |
| | dev-mm | 81.55 | **84.56** |
| | HANS | 54.25 | **67.11** |
| 200K | dev-m | 82.99 | **84.65** |
| | dev-mm | 83.41 | **85.04** |
| | HANS | 56.17 | **69.11** |
| 393K | dev-m | 83.87 | **85.17** |
| | dev-mm | 84.11 | **85.05** |
| | HANS | 61.22 | **71.67** |

Table 8: Results on MNLI when using different sizes of original training samples.

| Data Size | MNLI | | |
|---|---|---|---|
| | dev-m | dev-mm | HANS |
| 0K | 83.87 | 84.11 | 61.22 |
| 10K | 84.68 | 84.88 | 67.11 |
| 100K | 84.73 | 84.96 | 68.03 |
| 300K | 85.11 | **85.13** | 70.21 |
| 600K | 85.17 | 85.05 | **71.67** |
| 900K | **85.24** | 85.03 | 71.60 |
| $z$-filter | 82.55 | 82.70 | 67.69 |

Table 9: Results on MNLI when using different sizes of augmented datasets.

| Data | MNLI | | |
|---|---|---|---|
| | dev-m | dev-mm | HANS |
| IBADR (GPT-2 Large) | 85.17 | 85.05 | 71.67 |
| IBADR (LLaMA-7b) | **85.64** | **85.81** | **72.78** |

Table 10: Results of IBADR with GPT-2 Large and LLaMA-7b on MNLI.

randomly select two subsets from original training samples of MNLI, with sizes 100K and 200K, respectively. Afterwards, we employ IBADR to augment these subsets and subsequently retrain NLU models using the augmented datasets.

Table 8 presents the results of both the development and challenge sets. We can observe that IBADR consistently improves model performance across all test sets. Notably, even when using a limited number of original training samples, e.g. 100K, the model trained on the IBADR augmented dataset outperforms the full-size baseline. This suggests that IBADR exhibits remarkable robustness on limited original training samples.

### 3.9 The Effect of Augmented Dataset Size

To explore the influence of augmented dataset size, we retrain the NLU model on the MNLI dataset with different numbers of augmented samples: 10K, 100K, 300K, 600K, and 900K, respectively. As indicated in Table 9, the performance of IBADR consistently improves. Moreover, with just 100K augmented samples, IBADR outperforms $z$-filter across dev-m, dev-mm, and HANS. It's worth mentioning that $z$-filter utilizes a larger set of 360K augmented samples.

### 3.10 The Compatibility with Advanced Language Models

To ensure a fair comparison with $z$-filter, we employ GPT-2 Large as the sample generator in our main study. In exploring IBADR's compatibility with advanced large language models (LLM), we finetune the LLaMA-7b model (Touvron et al., 2023) using LORA (Hu et al., 2021) as an alternative to GPT-2 Large. The results on the MNLI

dataset are listed in Table 10. We can observe that the performance of IBADR is further improved with LLaMA-7b, which indicates IBADR's generalizability.

## 4 Related Work

Our related work primarily focuses on two categories of methods: model-centric and dataset refinement methods.

### 4.1 Model-centric Data Debiasing Methods

Numerous previous studies have adopted model-centric approaches to address biases in NLU models. Informed by their deep understanding of task-specific biases, they introduce innovative model architectures and training objectives aimed at preventing models from exploiting these biases. For instance, Belinkov et al. (2019) introduce an adversarial architecture specifically designed to mitigate hypothesis-only bias (Gururangan et al., 2018), while Stacey et al. (2020) enhance debiasing by employing multiple adversarial classifiers. Furthermore, there exists a complementary line of research focus on debiasing the model by down-weighting the importance of biased samples during training. The typical methods include example re-weighting (Reweight) (Schuster et al., 2019a), confidence regularization (Conf-reg) (Utama et al., 2020a; Du

et al., 2021), and products-of-experts (POE) (Clark et al., 2019; He et al., 2019; Karimi Mahabadi et al., 2020). Typically, these methods follow a two-stage paradigm. In the first stage, a bias-only model is trained, either automatically (Utama et al., 2020c; Geirhos et al., 2020; Sanh et al., 2021) or by leveraging prior knowledge about the bias (Clark et al., 2019; He et al., 2019; Belinkov et al., 2019). Then, in the second stage, the output of the bias-only model is utilized to adjust the loss function of the debiased model. Recently, Lyu et al. (2023) propose a novel approach using contrastive learning to capture the dynamic influence of biases and effectively reduce biased features, offering an alternative perspective on addressing bias in NLU models. Wang et al. (2023) observe that lower layers in Transformer models tend to capture biased features. They introduce the residual connection to integrate low-layer representations with top-layer ones, thus minimizing biased feature impact on the top layer.

### 4.2 Dataset Refinement

Several studies have explored generative data augmentation methods to enhance the model robustness in various domains. Lee et al. (2021) train a generator to generate new claims and evidence for debiasing fact verification datasets like FEVER. Ross et al. (2022) introduce TAILOR, a semantically-controlled perturbation method for data augmentation based on some manually defined perturbation strategies. Wu et al. (2022) identify a set of biased features by $z$-statistics, and then adjust the distribution of the generated samples by post-hoc filtering to remove the generated samples with biased features.

Unlike these approaches, our framework does not require data analysis to define biased features or manual perturbation rules, and hence achieves better generalizability.

## 5 Conclusions

In this work, we propose IBADR, an iterative dataset refinement framework for debiasing NLU models. Under this framework, we train a shallow model to quantify the bias degree of samples, and then iteratively generate pseudo samples with fewer biased features, which can be used to debias the model via retraining. We also incorporated two strategies to enhance the diversity of generated pseudo samples, further improving model performance. On extensive experiments of two tasks,

IBADR consistently shows superior performance compared to baseline methods. Besides, IBADR can better handle unknown biased features and has good compatibility with larger language models.

In the future, we will explore the compatibility of IBADR with other large language models, such as GPT4 (OpenAI, 2023).

## Limitations

The limitations of this framework are the following aspects: (i) Despite filtering the pseudo samples with low model confidence, IBADR might still produce pseudo samples with incorrect labels, which limits the model performance; (ii) We only conduct experiments on NLU tasks, neglecting the exploration of its applicability to a wider range of tasks.

## Ethics Statement

This paper proposes a dataset refinement framework that aims to adjust dataset distributions in order to mitigate data bias. All the datasets used in this paper are publicly available and widely adopted by researchers to test the performance of debiasing frameworks. Additionally, this paper does not involve any data collection or release, thus eliminating any privacy concerns. Oveall, this study will not pose any ethical issues.

## Acknowledgements

We would like to thank the anonymous reviewers for their insightful comments and suggestions. This research is supported by National Natural Science Foundation of China (No. 62276219) and Natural Science Foundation of Fujian Province of China (No. 2020J06001).

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
