# OpenReview forum: "IBADR: an Iterative Bias-Aware Dataset Refinement Framework for Debiasing NLU models"
_EMNLP/2023/Conference — EMNLP 2023 Main_

### Official Review · Reviewer_XyJx · 2023-08-03

**Soundness:** 4

**Excitement:**

3: Ambivalent: It has merits (e.g., it reports state-of-the-art results, the idea is nice), but there are key weaknesses (e.g., it describes incremental work), and it can significantly benefit from another round of revision. However, I won't object to accepting it if my co-reviewers champion it.

**Paper Topic And Main Contributions:**

This paper concentrates on the dataset refinement methods to debias the NLU models and improve their generalization under out-of-distribution (OOD) settings. It proposes an iterative data refinement framework for NLU. In each iteration, a shallow model is trained on a subset of the training data to measure the bias degree without manually-defined biased features. Then a generator, trained on samples paired with their bias degree, is instructed to generate pseudo samples with fewer biased features, expanding the training sets.

-------------Post-rebuttal---------------

I have read the author's response, and I think they have addressed my concerns. Therefore, I will improve my score.

**Reasons To Accept:**

- The proposed data refinement framework is simple, and does not rely on pre-defined types of biases.
- Good experimental results.
- The paper is well-organized and reader-friendly.

**Reasons To Reject:**

- The authors claim that the proposed method significantly outperforms the other baselines. However, no significant tests are performed.
- The experimental results on MNLI-hard and SNLI-hard (two widely-used OOD datasets for NLI) are lacking.
- The computational cost is high due to the iterative nature of the refinement process. Each iteration involves expanding the sample pool and retraining the generator, contributing to the overall expense of the framework.

**Reproducibility:**

4: Could mostly reproduce the results, but there may be some variation because of sample variance or minor variations in their interpretation of the protocol or method.

**Reviewer Confidence:**

4: Quite sure. I tried to check the important points carefully. It's unlikely, though conceivable, that I missed something that should affect my ratings.

---

> ### Author Rebuttal · Authors · 2023-08-28
>
> Thank you for reviewing our paper and for the insightful comments. We hope our answers to the 3 reject reasons will address the concerns and clarify the contributions of the paper.
>
> **R1: The authors claim that the proposed method significantly outperforms the other baselines. However, no significant tests are performed.**
>
> A1: When comparing IBADR and z-filter, we conduct the significance tests on HANS, MNLI-m hard and MNLI-mm hard, where the corresponding p-values are 0.0064, 0.0017, and 0.0031 respectively. This indicates that our framework is significantly better than z-filter across all three test sets. We will add this in our revised version.
>
> **R2: The experimental results on MNLI-hard and SNLI-hard (two widely-used OOD datasets for NLI) are lacking.**
>
> A2: Following the experimental setup of Utama et al., (2020)[1] and Lyu et al., (2023)[2], where HANS and scramble test sets are used to assess the debiasing effects of models on the MNLI and SNLI datasets. In response to your recommendation, we also conduct experiments on MNLI-hard and SNLI-hard. The average results across five runs with different random seeds are reported below. Obviously, our framework is also effective on these two datasets. Meanwhile, we conduct significance tests to investigate whether the performance improvement of IBADR over z-filter is significant.
>
> |              | SNLI-hard | MNLI-m hard | MNLI-mm hard |
> |--------------|:---------:|:-----------:|:------------:|
> | BERT-base    |   80.34   |    75.88    |     75.75    |
> | z-filter     |   82.82   |    79.19    |     78.51    |
> | IBADR        |   **83.43**   |    **79.62**    |     **80.01**    |
> | $p$-values    |   0.0052  |   0.0017   |     0.0031   |
>
> **R3: The computational cost is high due to the iterative nature of the refinement process. Each iteration involves expanding the sample pool and retraining the generator, contributing to the overall expense of the framework.**
>
> A3: Similar to prevalent data augmentation methods (e.g., Wu et al., 2022[3]; Lee et al., 2021[4]), the data augmentation process in IBADR is performed one-time in an offline manner. Once the augmented data is generated, it can serve as a valuable resource for training diverse downstream models.
>
> **References:**
> 1.Utama P A, Moosavi N S, Gurevych I. Towards debiasing NLU models from unknown biases. EMNLP 2020.
> 2.Lyu Y, Li P, Yang Y, et al. Feature-level debiased natural language understanding. AAAI 2023.
> 3.Wu Y, Gardner M, Stenetorp P, et al. Generating data to mitigate spurious correlations in natural language inference datasets. ACL 2022.
> 4.Lee M, Won S, Kim J, et al. Crossaug: A contrastive data augmentation method for debiasing fact verification models. CIKM 2021.

---

### Official Review · Reviewer_HAyt · 2023-08-04

**Soundness:** 3

**Excitement:**

4: Strong: This paper deepens the understanding of some phenomenon or lowers the barriers to an existing research direction.

**Paper Topic And Main Contributions:**

This paper proposes a new dataset refinement framework, which does not require human supervision about biases in the dataset, to debias NLU models. The main contributions of the paper are in three folds: 1) reducing the costs of human annotating, 2) may capturing overlooked biases, and 3) outperforming state-of-the-art by a large margin.

**Questions For The Authors:**

1.	What is the performance variance across the five trials?
2.	How does performance change with the number of augmented data? And at what point does it outperform existing bias reduction methods?
3.	Sanh et al., 2020 show that with a small model, it is possible to train a sufficiently biased model even using full data. But why did you train a shallow model with some datasets instead of the entire dataset?
4.	GPT-2 is a very old model. Why did you use that model despite the fact that various generative models with better performances was recently released?


**Reasons To Accept:**

The paper addresses an important problem of existing dataset refinement debiasing methods, such as the human cost problem, by an iterative dataset augmentation approach. This approach is simple and intuitive.

Experiment results in the paper show that the proposed method, IBADR, significantly improves the out-of-distribution performance of the pre-trained language model on various debiasing benchmarks. In particular, unlike most previous studies, IBADR improves both in-distribution performance and OOD performance simultaneously.

Though some moderately unclear claims exist, such as on lines 55-61, the paper is generally clear and well-written.


**Reasons To Reject:**

Since they use a small subset of training examples for biasing a shallow model, which is highly sensitive to sampled examples, the authors need to provide some analysis and experimental results on the robustness of their framework.

The authors use much more augmented data than the original data to achieve the performance, thus they need to show how much additional data is needed to achieve comparable performance with existing debiasing methods.


**Reproducibility:**

4: Could mostly reproduce the results, but there may be some variation because of sample variance or minor variations in their interpretation of the protocol or method.

**Reviewer Confidence:**

4: Quite sure. I tried to check the important points carefully. It's unlikely, though conceivable, that I missed something that should affect my ratings.

---

> ### Author Rebuttal · Authors · 2023-08-28
>
> Thank you for reviewing our paper and for the insightful comments. We hope our answers to the 4 questions will address the concerns and clarify the contributions of the paper.
>
> **Q1: What is the performance variance across the five trials?**
>
> A1: Taking the MNLI dataset as an example, we try five different random seeds for the generator, shallow model, and BERT-base. Throughout the experimentation process, we observe fluctuations in the shallow model's performance on the MNLI dev-m, spanning a range of 45% to 51%. However, we also discover that these fluctuations in the shallow model do not affect the quality of samples generated by IBADR. The variances in its performance on dev-m, dev-mm, and HANS are 0.05, 0.03, and 0.17 respectively.
>
> **Q2: How does performance change with the number of augmented data? And at what point does it outperform existing bias reduction methods?**
>
> A2:  To explore the influence of augmented dataset size, we retrain the NLU model on the MNLI dataset with different numbers of augmented samples: 10K, 100K, 300K, 600K, and 900K, respectively. The average results of 5 runs experiments are reported below. Please note that the original MNLI dataset contains 393K samples, and we report the model performance with additional 600K augmented samples in the original paper. From the following table, we can observe that the performance of our model can be always improved.
>
> |                       |   dev-m   |   dev-mm  |    HANS   |
> |-----------------------|:---------:|:---------:|:---------:|
> | 0K (from the paper)   |   83.87   |   84.11   |   61.22   |
> | 10K                   |   84.68   |   84.88   |   67.11   |
> | 100K                  |   84.73   |   84.96   |   68.03   |
> | 300K                  |   85.11   | **85.13** |   70.21   |
> | 600K (from the paper) |   85.17   |   85.05   | **71.67** |
> | 900K                  | **85.24** |   85.03   |   71.60   |
> | z-filter (360K)       |   82.55   |   82.70   |   67.69   |
>
> As shown in the above table, IBADR can surpass z-filter across dev-m, dev-mm, and HANS with only 100K augmented samples. It is noteworthy that z-filter[1] involves a larger set of 360K augmented samples.
>
> **Q3: Sanh et al., 2020 show that with a small model, it is possible to train a sufficiently biased model even using full data. But why did you train a shallow model with some datasets instead of the entire dataset?**
>
> A3: Previous studies have highlighted two primary approaches to construct biased models without manual intervention: 1. employing a reduced subset of dataset to train a biased model with the same architecture as the original one (Utama et al., 2020)[2]; 2. immobilizing a significant portion of parameters and training the model using the complete dataset (Sanh et al., 2020). Importantly, there is currently no work discussing the pros and cons of these two methods, which is also not our emphasis. For simplicity, we directly adopt the Utama's approach.
>
> To investigate the generality of our framework, we also follow Sanh et al., (2020) to retrain the shallow model and then reconduct the experiment on MNLI. The average results of 5 runs experiments are reported below.
>
> |                            | dev-m     | dev-mm    | HANS      |
> |----------------------------|-----------|-----------|-----------|
> | BERT-base                  | 83.87     | 84.11     | 61.22     |
> | z-filter                   | 82.55     | 82.70     | 67.69     |
> | IBADR (Utama et al., 2020) | 85.17     | **85.05** | **71.67** |
> | IBADR (Sanh et al., 2020)  | **85.31** | 84.94     | 71.41     |
>
> **Q4: GPT-2 is a very old model. Why did you use that model despite the fact that various generative models with better performances was recently released?**
>
> A4: To ensure a fair comparison with z-filter, we also use GPT2-large as the pretrained language model. Besides, since the sample generator is expected to model the co-occurrence relationships of bias indicators, labels, and sentences, we have to finetune the language model. However, finetuning large language models (LLMs) poses challenge due to cost or feasibility issues, especially in cases of commercial LLMs or GPT3 via the OpenAI API. As a workaround, researchers have proposed some finetuning-efficient techniques, such as LORA. Based on this, we explore finetuning llama-7B with the help of LORA, as a replacement for GPT2-large. The results are reported below.
>
> |                  | dev-m | dev-mm | HANS  |
> |------------------|-------|--------|-------|
> | IBADR (GPT-2)    | 85.17 | 85.05  | 71.67 |
> | IBADR (llama-7B) | **85.64** | **85.81**  | **72.78** |
>
> **References:**
>
> 1.Wu Y, Gardner M, Stenetorp P, et al. Generating data to mitigate spurious correlations in natural language inference datasets. ACL 2022.
>
> 2.Utama P A, Moosavi N S, Gurevych I. Towards debiasing NLU models from unknown biases. EMNLP 2020.

---

### Official Review · Reviewer_NJeA · 2023-08-05

**Soundness:** 4

**Excitement:**

3: Ambivalent: It has merits (e.g., it reports state-of-the-art results, the idea is nice), but there are key weaknesses (e.g., it describes incremental work), and it can significantly benefit from another round of revision. However, I won't object to accepting it if my co-reviewers champion it.

**Paper Topic And Main Contributions:**

This paper considers the dataset biases problem that impedes the model generalization ability to out-of-distribution data. There are two mainstreams to mitigate the detrimental effect of the biased samples, which are model-centric and dataset-refinement approaches. Among them, the proposed method is built on the dataset refinement approach as it does not require modification on model architectures and training objectives. Unlike the previous studies that rely on the human-annotated bias dataset, the proposed method, IBADR, iteratively refines the training dataset such that the remaining dataset only includes the bias-conflicting samples. Specifically, IBADR adopts the shallow model to identify the bias samples and fine-tunes LMs to only generate bias-conflicting samples. The above processes are iteratively repeated, and the final samples are only used to train the main networks.

As a review's viewpoint, the contribution of this paper are as follows:

(i) introducing the first dataset-refinement approach which does not require human-annotated bias labels.

(ii) better debiasing performance compared to existing two approaches.

**Questions For The Authors:**

Q1) How sensitive the model is on the different number of augmented dataset?

Q2) Do the better auto-regressive models (e.g., GPT-3) can improve the debiasing performance?

Q3) Why do the authors use the same model (i.e., BERT-base) for both the main and shallow models in the experiment of Table 3, which is not consistent to the explanation?

Q4) Did the authors compare the proposed method with existing and simple augmentation techniques (other than CrossAug)? because there is possibility that two time more augmented dataset can just simply mitigate the dataset bias.

Q5) Are there any failure cases that the augmented text are not aligned with the given labels?

**Reasons To Accept:**

As I stated in the contribution parts, the strengths of this paper are as follows:

(i) a new method for the dataset-refinement approach that does not require human-annotated biases.

(ii) the proposed dataset-refinement methods shows better OOD and ID performance compared to model-centric and existing dataset-refinement approaches.



**Reasons To Reject:**

The weakness of this paper is the weak analysis about the sample generator. While utilizing sample generator leads to better performance, the detailed analysis on the sample generators are absent. For example, (i) the qualitative analysis on the generated examples (ii) the effect of the different number of augmented samples. This is crucial because the current version requires at least 2 time (approximately 3 times) more training dataset than the original dataset.

**Reproducibility:**

4: Could mostly reproduce the results, but there may be some variation because of sample variance or minor variations in their interpretation of the protocol or method.

**Reviewer Confidence:**

3: Pretty sure, but there's a chance I missed something. Although I have a good feel for this area in general, I did not carefully check the paper's details, e.g., the math, experimental design, or novelty.

---

> ### Author Rebuttal · Authors · 2023-08-29
>
> Thank you for reviewing our paper and for the insightful comments. We hope our answers to the 5 questions will address the concerns and clarify the contributions of the paper.
>
> **Q1: How sensitive the model is on the different number of augmented dataset?**
>
> To explore the influence of augmented dataset size, we retrain the NLU model on the MNLI dataset with different numbers of augmented samples: 10K, 100K, 300K, 600K, and 900K, respectively. The average results of 5 runs of experiments are reported below. Please note that the original MNLI dataset contains 393K samples, and we report the model performance with additional 600K augmented samples in the original paper. From the following table, we can observe that the performance of our model can be always improved.
>
> |                       |   dev-m   |   dev-mm  |    HANS   |
> |-----------------------|:---------:|:---------:|:---------:|
> | 0K (from the paper)   |   83.87   |   84.11   |   61.22   |
> | 10K                   |   84.68   |   84.88   |   67.11   |
> | 100K                  |   84.73   |   84.96   |   68.03   |
> | 300K                  |   85.11   | **85.13** |   70.21   |
> | 600K (from the paper) |   85.17   |   85.05   | **71.67** |
> | 900K                  | **85.24** |   85.03   |   71.60   |
> | z-filter (360K)       |   82.55   |   82.70   |   67.69   |
>
> **Q2: Do the better auto-regressive models (e.g., GPT-3) can improve the debiasing performance?**
>
> To ensure a fair comparison with z-filter, we also use GPT2-large as the pretrained language model. Besides, since the sample generator is expected to model the co-occurrence relationships of bias indicators, labels, and sentences, we have to finetune the language model. However, finetuning large language models (LLMs) poses challenge due to cost or feasibility issues, especially in cases of commercial LLMs or GPT3 via the OpenAI API. As a workaround, researchers have proposed some finetuning-efficient techniques, such as LORA. Based on this, we explore finetuning llama-7B with the help of LORA, as a replacement for GPT2-large. The results are reported below.
>
> |                  | dev-m | dev-mm | HANS  |
> |------------------|-------|--------|-------|
> | IBADR (GPT-2)    | 85.17 | 85.05  | 71.67 |
> | IBADR (llama-7B) | **85.64** | **85.81**  | **72.78** |
>
> **Q3: Why do the authors use the same model (i.e., BERT-base) for both the main and shallow models in the experiment of Table 3, which is not consistent to the explanation?**
>
> As mentioned in Lines 149-153, we adopt the commonly-used approach proposed by Utama et al., 2020[1] to train the shallow model. Note that this shallow model shares the same architecture as the main model, and is trained with only a small subset of the training data.
>
> **Q4: Did the authors compare the proposed method with existing and simple augmentation techniques (other than CrossAug)? because there is possibility that two time more augmented dataset can just simply mitigate the dataset bias.**
>
> As shown in Table 6, we compare IBADR with simple augmentation techniques, including text swapping, paraphrasing and synonym substitution. Experimental results demonstrate the effectiveness of IBADR. Notably, as mentioned in the response to Q1, IBADR can maintains its effectiveness even with only 10K augmented samples.
>
> **Q5: Are there any failure cases that the augmented text are not aligned with the given labels?**
>
> Yes, we have investigated the quality of the augmented samples. Specifically, we randomly select 100 samples from the augmented MNLI dataset and engage 3 human annotators to manually classify the relation between sentence pairs. We apply the majority vote to finalize annotations. The accuracy of augmented instances is about 92% based on human-annotated gold labels. We will add these details in our revised version.
>
> **References:**
>
> 1.Utama P A, Moosavi N S, Gurevych I. Towards debiasing NLU models from unknown biases. EMNLP 2020.

---

### Meta-Review · Area_Chair_7AbG · 2023-09-17

**Recommendation:** 4

**Metareview:**

This paper presents the Iterative Bias-Aware Dataset Refinement (IBADR) framework, a new dataset-refinement method for debiasing NLU models. The authors introduce an iterative process involving a shallow model and a sample generator. The shallow model identifies biases in samples while the generator produces pseudo samples with fewer biased features in each iteration. Experimental results show that IBADR not only outperforms existing dataset refinement methods but also surpasses performance from model-centric approaches. However, potential weaknesses in this study include substantial data augmentation requirements and lack of comprehensive analysis surrounding the implemented sample generator.

The key novelty of this approach is its independence from predefined biased features or manual data analysis, which contributes to overcoming crucial limitations in contemporary dataset refinement approaches. Comprising a shallow model and a sample generator, the IBADR framework effectively identifies bias degrees of samples and generates pseudo samples with fewer biased features iteratively.

Some reviewers pointed out that the paper falls short in providing an exhaustive analysis on the role and impact of the sample generator. They also pointed out the lack of significance testing, computational costs and the heavy reliance on augmented samples raising questions over scalability. The authors have offered adequate analysis in author response, which has been acknowledged by the reviewers to be satisfactory.

---

### Decision · Program_Chairs · 2023-10-07

**Decision:**

Accept-Main

**Comment:**

This paper presents the Iterative Bias-Aware Dataset Refinement (IBADR) framework, a new dataset-refinement method for debiasing NLU models. The authors introduce an iterative process involving a shallow model and a sample generator. The shallow model identifies biases in samples while the generator produces pseudo samples with fewer biased features in each iteration. Experimental results show that IBADR not only outperforms existing dataset refinement methods but also surpasses performance from model-centric approaches. However, potential weaknesses in this study include substantial data augmentation requirements and lack of comprehensive analysis surrounding the implemented sample generator.

The key novelty of this approach is its independence from predefined biased features or manual data analysis, which contributes to overcoming crucial limitations in contemporary dataset refinement approaches. Comprising a shallow model and a sample generator, the IBADR framework effectively identifies bias degrees of samples and generates pseudo samples with fewer biased features iteratively.

Some reviewers pointed out that the paper falls short in providing an exhaustive analysis on the role and impact of the sample generator. They also pointed out the lack of significance testing, computational costs and the heavy reliance on augmented samples raising questions over scalability. The authors have offered adequate analysis in author response, which has been acknowledged by the reviewers to be satisfactory.